# Characterization of a Commercial Whey Protein Hydrolysate and Its Use as a Binding Agent in the Whey Protein Isolate Agglomeration Process

**DOI:** 10.3390/foods11121797

**Published:** 2022-06-18

**Authors:** Baheeja J. Zaitoun, Niels Palmer, Jayendra K. Amamcharla

**Affiliations:** 1Department of Animal Sciences and Industry, Food Science Institute, Kansas State University, Manhattan, KS 66506, USA; bzaitoun@ksu.edu; 2Glanbia Nutritionals, Twin Falls, ID 83301, USA; npalmer@glanbia.com

**Keywords:** agglomeration, whey, hydrolysate, isolate

## Abstract

The first objective of this study was to characterize the chemical properties of three lots of whey protein hydrolysate (WPH) obtained from a commercial manufacturer. The degree of hydrolysis (DH) of WPH was between 13.82 and 15.35%, and was not significantly (*p* > 0.05) different between the batches. From MALDI-TOF, 10 to 13 different peptides were observed in the range of 2.5–5 kDa and 5–8 kDa, respectively. The second objective of the study was to evaluate the effectiveness of WPH as a binder in whey protein isolate (WPI) wet agglomeration. For this purpose, a 3 × 3 × 2 factorial design was conducted with pre-wet mass (60, 100, and 140 g), WPH concentration (15, 20, and 25%), and flow rate (4.0 and 5.6 mL·min^−1^) as independent variables. WPI agglomeration was carried out in a top-spray fluid bed granulator (Midi-Glatt, Binzen, Germany). Agglomerated WPI samples were stored at 25 °C and analyzed for moisture content (MC), water activity, relative dissolution index (RDI), and emulsifying capacity. Pre-wet mass, flow rate, and the WPH concentration had a significant (*p* < 0.05) effect on the MC. Moreover, all interactions among the main effects had also a significant (*p* < 0.05) effect on MC. High MC and water activity were observed for the treatments with a higher pre-wet volume and higher flow rate, which also resulted in clumping of the powders. The treatment with the 60 g pre-wet mass, 20% WPH concentration, and 5.6 mL·min^−1^ flow rate combination had the highest RDI among all the samples. In conclusion, WPH can be used as a potential alternative to soy lecithin in WPI wet agglomeration.

## 1. Introduction

Whey is a co-product obtained during the manufacturing of cheese. It was considered a waste for decades until research studies showed its functionality and nutritional value. Sweet whey contains approximately 6.0 to 10 g/L of proteins [1]. The major proteins in whey include β-lactoglobulin (β-LG), α-lactalbumin (α-LA), bovine serum albumin (BSA), and immunoglobulins (Ig) [2]. Whey proteins (WPs) are highly soluble and have unique physicochemical characteristics that influence their functionality in food applications such as gelation, emulsification, and foaming [3]. This makes it a feasible ingredient in various food products. However, some approaches are performed to modify the functional and physicochemical properties of WPs, such as chemical, physical, and enzymatic treatments. Most research on protein modification includes Maillard conjugation [4], physical modification such as high-pressure treatment [5], and enzymatic modifications [6]. Modification can also be carried out by applying enzymatic and physical treatments [7]. Enzymatic hydrolysis of WPs is measured by the degree of hydrolysis, and it is defined as the percentage of peptide bonds cleaved [8]. A low DH (<10%) is sufficient for improving the physicochemical properties of WPs, whereas, a DH (>10%) is more suitable for improving the biological functions of the resultant peptides such as antimicrobial, antioxidant, antihypertensive, and immunomodulatory functions [9,10]. Enzymatic hydrolysis of WPs was extensively studied [11]. It was proven that hydrolysis improves WP digestibility and nutritional value and reduces allergenicity [9], making it a suitable ingredient in infant formula [12]. Hydrolysis also enhances the solubility [13] and the emulsifying capacity at an alkaline pH of the whey hydrolysate [14,15]. In a recent disclosure of the invention, Palmer et al. [16] found that the hydrolysate worked as an excellent binder in the WPI agglomeration process as an alternative to soy lecithin.

Agglomeration improves the reconstitution properties of the powders due to the incorporation of air between powder particles, which makes the water penetration into these particles easier during subsequent rehydration. Therefore, the agglomerates readily disperse and dissolve quickly [17] compared to non-agglomerated powders. More studies are needed to better understand the physical and chemical properties of this whey protein hydrolysate. In addition, the agglomeration process conditions need to be optimized, and their effects on the resultant powder properties studied. Therefore, the first objective of this study is to characterize the physical and chemical properties of three lots of a commercial WPH. Subsequently, optimizing and evaluating the effectiveness of WPH as a binder in WPI wet agglomeration was investigated as a second objective.

## 2. Materials and Methods

### 2.1. Experimental Design

Three lots of WPH and one lot of WPI were obtained from a commercial manufacturer (Glanbia Nutritionals, Twin Falls, ID, USA). The batches were randomly selected and had a month’s difference between manufacturing dates to assure that these batches were completely independent and not made from the same whey batch. Initially, the chemical and physical properties of WPH and WPI were analyzed in terms of the peptide characterization, degree of hydrolysis, zeta potential, color, bulk, and tapped densities, to evaluate the consistency of the enzymatic hydrolysis. After determining the similarities and differences among WPH batches, the effectiveness of using WPH as a binder in WPI wet agglomeration was evaluated. For this purpose, a 3 × 3 × 2 factorial design was conducted with pre-wet mass (60, 100, and 140 g), WPH concentration (15, 20, and 25%), and flow rate (4.0 and 5.6 mL·min^−1^) as independent variables. The other processing parameters, such as the nozzle pressure, fluid bed pressure, and fluid bed temperature, were set at 0.65 bar (9.43 psi), 0.45 bar (6.53 psi), and 60 °C, respectively.

The pre-wet mass represents the weight of water (g) used in wetting the WPI powder as the first agglomeration step. WPH concentration is the concentration (*w*/*w*) of the binder solution. Lastly, the flow rate is the amount of water and binder solution pumped during agglomeration, expressed as mL.min-1. WPI agglomeration was performed in a top-spray fluid bed granulator (Midi-Glatt, Binzen, Germany), as shown in Figure 1. All the experiments were performed in triplicate using the three batches of WPH. Agglomeration was stopped when the temperature of the end powder reached 45 °C. Agglomerated WPI samples were stored at 25 °C and analyzed for moisture, water activity, relative dissolution index (RDI), and emulsifying capacity.

### 2.2. WPH and WPI Chemical Characterization

#### 2.2.1. HPLC and MALDI-TOF Mass Spectrometry

About 1 g of WPH powder was dissolved in distilled deionized water and dialyzed using 2 kDa cut-off cassettes (Slide-A-Lyzer^®^ Dialysis Cassettes, Thermo Scientific, Waltham, MA, USA). The dialyzed WPH solution was passed through 0.45 µm filter units (MF-Millipore™ membrane, Carrigtwohill Co., Carrigtwohill, Ireland). Then, protein concentrations in the dialyzed solution were determined by measuring the absorbance at 280 nm using a Cary 50 bio UV-visible spectrophotometer (Agilent Technologies, Santa Clara, CA, USA). Then, their concentrations were standardized by taking the number of tyrosine and tryptophan residues from α-LA, β-LG, and BSA into account with their extension coefficient [18]. Samples were stored at −18 °C until use.

HPLC analysis was carried out based on the method described by [19], with some modifications. The separation of whey proteins was performed using RP-HPLC at 25 °C with a Beckman Coulter System Gold^®^ (Beckman Coulter, Fullerton, CA, USA), with a 126 solvent module pump, 168 System Gold^®^ detector, and Beckman 32 Karat software data acquisition system. A ZORBAX 300SB-C8 column, 2.1 × 150 mm with 5 µm pore size, was used in the analysis (Agilent Technologies Co., Englewood, CO, USA). A 50 µL aliquot of a 20 µM sample was injected into the column. Solvent A of the mobile phase was 0.1% trifluoroacetic acid (TFA, American Bioanalytical Co., Natick, MA, USA) in 99.9% H_2_O, and solvent B was 0.1% TFA, 90% acetonitrile (Fisher Scientific-HPLC grade) and 9.9% H_2_O. The gradient of solvent B started at 10% for 5 min, then increased to 30% over 5 min. Subsequently, the gradient continuously increased to 80% over 30 min, then to 100% over 5 min. Lastly, the gradient was held at 100% for solvent B for an additional 10 min and then brought back down to 10% to re-equilibrate for the next run. The flow rate was 0.5 mL/min. When the peak appeared on the HPLC, samples were collected in Eppendorf tubes, and the peptides were analyzed on MALDI-TOF. The α-LA, β-LG, and BSA standards were purchased from Sigma-Aldrich (St. Louis, MO, USA). Detection was performed at a wavelength of 168–220 nm.

Ultraflex II MALDI-TOF (Burker Daltonics, Bremen, Germany) was used in analyzing the peptides from HPLC fractions [20]. A 20 mg/mL 2,5- Dihydroxybenzoic acid (DHB) matrix (Sigma Aldrich, St. Louis, MO, USA) was used in a solution of 50%. Samples were spotted using 1% TFA in acetonitrile for the analysis.

#### 2.2.2. Degree of Hydrolysis (DH)

The DH was measured using the 2,4,6 trinitrobenzene sulfonic acid (TNBS) method [8]. A 50 mL sample of a 0.02% protein solution was prepared by dissolving 1 g of WPH powder in distilled water. Protein solutions were stirred for 30 min to ensure that the powder was fully dissolved. Then, 1 g of the protein solution was added to 9 g of 1% (*w*/*w*) sodium dodecyl sulfate (SDS) solution (Sigma Aldrich, St. Louis, MO, USA). In test tubes, a 0.1 mL aliquot of protein solution, 2 mL of phosphate buffer (pH 8.2), and 2 mL of 0.1% (*w*/*w*) TNBS solution (Sigma Aldrich, St. Louis, MO, USA) were added. A leucine standard linear curve was obtained by diluting 530 mg/L leucine standard (Sigma Aldrich, St. Louis, MO, USA) in 1% SDS (*w*/*w*) to obtain 42.4, 84.8, 127.2, 169.6, and 212 mg/L amino nitrogen leucine standards. All tubes were vortexed and placed in the water bath at 50 °C for 60 min. The reaction was stopped by adding 4 mL of 0.1 N HCL (Chemicals, Gibbstown, NJ, USA) to each tube. Standards and samples were measured at 340 nm using a UV/VIS spectrophotometer (Metash Instruments Inc., Shanghai, China) against a 1% SDS blank solution. The leucine standard curve was plotted, then calculations were carried out accordingly.

### 2.3. WPH and WPI Physical Characterization

#### 2.3.1. Water Activity

The water activity of powders was measured at 25 °C using an Aqua Lab Series 3 model TE instrument (Pullman, WA, USA). The measurements were performed in triplicate. The same procedures were followed in analyzing the water activity of agglomerated powders.

#### 2.3.2. Mean Particle Size and Zeta Potential (ζ)

The particle size distribution and zeta potential were measured using dynamic light scattering (DelsaMax PRO, Beckman, Germany) for the reconstituted WPH or WPI solutions. Initially, 5 g of WPH or WPI was dissolved in 95 g of distilled water and stored in the fridge overnight for complete rehydration. The 5% solutions were diluted at 1:100 with distilled water and then slowly injected into the DLS flow cell. The measurements were performed in triplicate.

#### 2.3.3. Bulk and Tapped Densities

Bulk and tapped densities were measured by the Hosokawa Micron PT-R powder tester (Hosokawa Micron Corp., Osaka, Japan). A 100 cm^3^ cup and 180 vertical taps were applied to measure the tapped density. Standard steps were used in conducting both of the analyses. The measurements were performed in triplicate.

#### 2.3.4. Color

The CIE LAB values L*, a*, and b* were measured using a Hunter-Lab Mini Scan colorimeter (Hunter Associates Laboratory, Reston, VA, USA). Approximately 2 g of powder was placed in a transparent plastic container to take the measurement. The analysis was performed in triplicate. The L* value indicates the whiteness/darkness of the sample; a* indicates the redness/greenness, and b* shows the yellowness/blueness of the powder.

### 2.4. The Use of WPH as a Binder in Agglomerating WPI

A top-spray fluid bed granulator (Midi-Glatt; Glatt Process Technology, Binzen, Germany) was used, with a peristaltic pump (Model 1B.1003-R/65, Petro Gas, Berlin, Germany) attached, as shown in Figure 1. A total of 480 g of WPI was weighed in the agglomerator chamber. The WPH solution was prepared at 25 °C, and two drops of the SUPPRESSOR 3569 defoamer (Hydrite Chemical Co., Brookfield, WI, USA) were added to prevent foam formation. Processing parameters such as the pre-wet mass (60, 100, and 140 g), WPH concertation (15, 20, and 25%), and flow rate (4.0, 5.6 mL·min^−1^) were set according to the particular variable combination for each trial. The other processing parameters were fixed. The nozzle pressure, fluid bed pressure, and fluid bed temperature were set at 9.43 psi (0.65 bar), 6.53 psi (0.45 bar), and 60 °C, respectively.

Distilled water was added as a pre-wetting step, followed by inserting the WPH solution to the system. When 20 mL of the WPH solution remained, the fluid bed temperature was increased from 60 °C to 80 °C. The process was stopped when the temperature of the agglomerated powder reached 45 °C. Subsequently, samples were collected in plastic containers (48 oz, PET square grip container, Container and Packaging Co., Louisville, KY, USA) and stored at 25 °C. Samples were analyzed for moisture, water activity, solubility index, relative dissolution index, and emulsifying capacity. All the experiments were performed in triplicate using the three WPH batches.

### 2.5. Agglomerated WPI Characterization

#### 2.5.1. Moisture Content (MC)

The MC of agglomerated samples was determined using the direct forced oven following the AOAC official method 990.20 (AOAC, 2000). Aluminum pans were dried overnight at 102 °C then placed in a desiccator to cool down. Two grams of powder was weighed in a pan and placed in the direct forced oven (Fisher Scientific Isotemp oven 737G, Waltham, MA, USA) at 100 °C for 4 h. Analyses were performed in duplicate.

#### 2.5.2. Relative Dissolution Index (RDI)

The dissolution characteristics of the agglomerated WPI samples were evaluated using a focused beam reflectance measurement (FBRM), following the method proposed by Hauser and Amamcharla [21] with some modifications. In a 500 mL glass beaker, 5% (*w*/*w*) protein solutions of agglomerated WPI were prepared by dissolving powders in distilled water, while the temperature was maintained at 25 °C. The FBRM instrument is equipped with an overhead stirrer 4-blade impeller (Caframo, Georgian Bluffs, ON, Canada) that was rotating at 700 rpm during the powder addition (~40–45 s), then it was set at 400 rpm for data collection. The dissolution test was held for 30 min using the iC FBRM software (version 4.3.391, Mettler-Toledo AutoChem Inc., Columbia, MD, USA) to gather the data. The software program enabled the collection of the number of particles in the category of <10 μm chord length, that is, characterized as fine particles. The particle count was plotted against dissolution time. Subsequently, the area under the fine particle count curve was calculated to determine the dissolution using the trapezoidal rule. The RDI of the agglomerated powders was determined using Equation (1) below [22]:(1)$$\mathrm{RDI}\;(\%)=\frac{Area\;under\;the\;curve\;for\;the\;sample\;}{The\;highest\;area\;under\;the\;curve\;among\;agglomerated\;samples}\times 100\;$$

#### 2.5.3. Solubility Index (SI)

Twenty grams of the 5% (*w*/*w*) protein solutions that were prepared in the FBRM analysis were weighed into 50 mL centrifuge tubes. Samples were centrifuged at 700× *g* for 10 min at 25 °C in a Marathon 21000R centrifuge (Fisher Scientific, Pittsburgh, PA, USA). Aluminum pans were dried overnight at 102 °C and left in a desiccator to cool down. SI was determined based on the total solid of the supernatant. Three grams of the supernatant was weighed in the pan then placed in the oven at 100 °C for 4 h. Final weights were recorded. Subsequently, the amount of soluble material (𝜎) was calculated following Equation (2) below:(2)$$𝜎=\frac{Weight\;of\;dry\;material}{Weight\;of\;solution}\times 100\;$$

#### 2.5.4. Emulsifying Capacity (EC)

The EC of samples were determined by following the Webb et al. [23] method. At room temperature, 120 mL of 0.05% protein solution was prepared in a 600 mL beaker. The conductivity meter electrode (Accumet™ AP75; Fisher Scientific, Pittsburgh, PA, USA), and CAT Scientific X120 Handheld Homogenizer Drive (PolyScience, Niles, IL, USA) were placed in the beaker. With continuous blending, the addition of vegetable oil, that was purchased locally, started by opening the 125 mL separatory funnel valve all the way. The blending process was started at a slow rate (speed 1), then was increased to speed 3 when the emulsion became more viscous. Oil addition was stopped when the emulsion breakpoint was detected. This point is defined as the sudden increase in the electrical resistance of the dispersion that occurs upon emulsion collapse. The emulsification capacity was expressed as the total grams of oil emulsified per 100 mg of soluble protein [24].

### 2.6. Statistical Analysis

All data was analyzed based on the randomized complete block design (RCBD) using PROC GLIMMIX in SAS Studio (version 9.4; SAS Inst., Cary, NC, USA) and Tukey’s test to determine any significant differences between treatment levels, which were declared significant when *p* ≤ 0.05. WPH batches were the blocking factor in our experimental design. In addition, PROC RSREG was used to get the response surface models for moisture content and relative dissolution index. The general model equation is Ŷ = β_0_ + β_1_x_1_ + β_2_x_2_ + β_3_x_3_ + +β_11_(x_1_)^2^ + β_12_x_1_x_2_ + β_13_x_1_x_3_ + β_22_(x_2_)^2^ + β_23_x_2_x_3_, where β_0_, β_1_, β_2_, β_3_, β_11_, β_22_, β_12_, β_23_, and β_13_ are the constant coefficients, x_1_ is the pre-wet mass (g), x_2_ is the WPH concentration (*w*/*w*), and x_3_ is the flow rate (mL·min^−1^).

## 3. Results and Discussion

The chemical compositions of the WPI and WPH batches were provided by the manufacturer (Glanbia Nutritionals, Twin Falls, ID, USA). The moisture content of the WPI sample was 3.61%, whereas the WPH moisture content was within the range of 2.67–2.99% (Table 1). The WPI protein content was 94.02%, and the WPH protein content was within the range of 91.57–92.28%.

### 3.1. WPH Chemical Characterization

#### 3.1.1. HPLC and MALDI-TOF Mass Spectrometry

The standards of BSA, β-LG, and α-LA were identified on the HPLC and their peaks occurred at 26.3, 27.2, and 29.9 min, respectively. The major components of WPs (β-LG, α-LA, and BSA) were observed on the WPI-HPLC chromatogram. In addition, two unidentified peaks were also observed. However, none of those peaks were detected on the WPH sample chromatograms (Figure 2), which suggests the complete hydrolysis of these major whey proteins in the WPH batches. This finding was confirmed with the MALDI-TOF results. The MW of BSA, β-LG and α-LA are approximately 69, 18.3, and 14 kDa, as was noted by Goodall et al. [2]. The peptides on the MALDI-TOF spectrum were classified as small (2.5–5 kDa), medium (6–10 kDa), and large (>10 kDa). In the WPI sample, approximately 25% of the peptides found were small peptides, which was the same for the medium size ones. The remaining 50% of the peptides were classified as large peptides. These proteins had a MW of 18.3 and 14 kDa, which are the β-LG, and α-LA WPs. In WPH samples, about 75% of the peptides that were observed were classified as small peptides and 25% as medium peptides. In addition, no peptides were observed above 10 kDa, which confirms the HPLC findings that the major whey proteins were completely hydrolyzed. Moreover, the largest MW of a peptide observed in all WPH samples was 8 kDa. These results were similar to the findings reported by Adjonu et al. [25]. All peptides were <8 kDa and 52% of the obtained peptides were in the range of 2.1–8.0 kDa when chymotrypsin was used as an enzyme after 24 h of hydrolysis. The HPLC chromatograms for all WPH lots were similar. No peaks were observed at the times of 26.3, 27.2, and 29.9 min, indicating a complete hydrolysis of the major WPs in WPH samples. This finding was confirmed with the MALDI-TOF results, as the major WP peptides were detected in the WPI samples but not in the WPH samples. Both HPLC and MALDI-TOF results showed that the major WPs were completely hydrolyzed, indicating a consistent hydrolysis.

#### 3.1.2. Degree of Hydrolysis (DH)

It was noted by Spellman et al. [26] that the TNBS method is the most suitable for quantifying the DH in WPH, compared with the o-phthaldialdehyde (OPA) and pH-stat methods. TNBS reacts with amino groups, forming a chromophore with a maximum absorbance at 340 nm [8]. The DH of the WPH samples was between 13.82 and 15.35%, and was not significantly (*p* > 0.05) different between the batches (Table 1). Limited hydrolysis improves the physio-functional properties of the hydrolysate, as it was concluded by Lieske and Konrad [27] that the optimal DH is 3% to improve the foaming and emulsifying capacity. Partial hydrolysis improved the thermal stability of WPH due to the loss of the secondary structure [3]. Moreover, a DH > 10% is more suitable for improving the biological functions of the resultant peptides, such as antimicrobial, antioxidant, antihypertensive, and immunomodulatory functions [7,8]. Adjonu et al. [25] had used the OPA method in measuring the DH and reported that WP hydrolysis had significantly increased the biological functions of WPs, such as oxygen radical absorbance capacity and ACE-inhibition activity. However, extended hydrolysis (up to 24 h) had no significant effect on the DH and the molecular weight profiles (*p* > 0.05), but in some instances caused a reduction in the antioxidant activity of WPI hydrolysates.

### 3.2. WPH and WPI Physical Characterization

#### 3.2.1. Water Activity

Water activity of WPH samples was within the range of 0.16–0.21 (Table 2). The water activity of the WPH samples was higher than that of the WPI sample due to the differences in the chemical compositions such as protein content and moisture content. For example, WPI contained 94.02% proteins, compared with about 91.57% in lot 2 of the WPH. The higher protein content resulted in a lower water activity. There was a significant difference (*p* < 0.05) between lot 2 and 3 samples.

#### 3.2.2. Mean Particle Size and Zeta Potential (ζ)

Mean particle size and zeta potential were measured for the rehydrated samples. The mean particle size was within the range of 150.67–198.93 µm, and it was significantly different (*p* < 0.05) among the three WPH batches (Table 2). The mean particle size of WPH samples was bigger than those of the WPI samples due to the aggregation behavior of the hydrolysate. The hydrolysate may cause aggregation but not necessarily induce gelation, as Spellman et al. [26] noted. Zeta potential is an indicator of the surface charges of the particles. The mean zeta potential ranged from −22.88 to −24.04 mV, and there was no significant difference (*p* > 0.05) among the WPH batches (Table 2).

Zeta potential of the hydrolysate samples was higher than the intact WPI. This is due to the increase in the net charge on the protein hydrolysate [28]. The increase in negative charges might be due to the increase in the number of hydrophobic amino acid side chains after enzymatic hydrolysis [29]. In addition, as the absolute value of zeta potential increases, the stability of dispersion increases [30]. Both egg white and whey proteins are globular proteins, which explains why they both had similar zeta potential results when hydrolyzed.

#### 3.2.3. Bulk and Tapped Densities (g/cm^3^)

The average bulk density values of WPH samples was within the range of 0.31–0.33 g/cm^3^ (Table 3). Additionally, the mean tapped density was in the range of 0.42–0.45 g/cm^3^. WPH powders were statistically different (*p* < 0.05) in bulk and tapped densities. However, these values are similar practically. WPH samples had slightly higher bulk and tapped densities than the intact WPI densities. This is due to the lower volume of occluded air in the WPH samples [31].

#### 3.2.4. Color

The means of the L*, a*, and b* values were used to evaluate the differences in color intensity among the WPI and WPH samples. The average of L* was 92.15 and there was no significant difference (*p* > 0.05) among the three WPH lots in the L* value (Table 3). Two samples had similar a* and b* values with a mean of 0.15 and 7.39, respectively. However, the third batch was significantly different (*p* < 0.05) from the other two batches, with a* and b* values of 1.54 and 10.52, respectively. This variation might have been caused by some minor manufacturing variation such as the degree of hydrolysis and/or moisture content. Lot 3 had the lowest moisture content and degree of hydrolysis among WPH lots. These differences are very minor and do not have any effect on the use of these batches as a binding agent in the agglomeration process.

### 3.3. Agglomerated WPI Characterization

Wet agglomeration involves spraying a liquid binder on the powder in a fluidized bed chamber, causing the adhesion of wet particles due to viscous bridges between the particles. With the continuous supply of hot air from the fluid bed, the viscous bridges consolidate. The resultant agglomerates have a porous structure that improves the dissolution rate and flowability, and decreases the apparent bulk density [32]. In this study, agglomerated powders were tested for their moisture content, water activity, solubility index, relative dissolution index (RDI), and emulsifying capacity.

#### 3.3.1. Moisture Content (MC)

According to the ADPI [33] standards, MC should not exceed 6% by wt in dry whey products. The MC for all treatments (Table 4) was in the normal range (<6%), except in treatments 14 (140 g, 15% and 5.6 mL·min^−1^), 16 (140 g, 20% and 5.6 mL·min^−1^), and 18 (140 g, 25% and 5.6 mL·min^−1^), which contained a MC of 14.79, 7.56, and 7.41%, respectively. In these treatments, clumps were formed (Figure 3). Therefore, monitoring the temperature of the end product was difficult, which resulted in a large variation of MC in these treatments. On the other hand, MC in the treatments that have the combination of a high moisture content (140 g) and low flow rate (4.0 mL·min^−1^) was between 3.30% and 3.56% (Table 4), which matches the standard requirements of dry whey products.

In treatments 2 (60 g, 15%, and 5.6 mL·min^−1^), 4 (60 g, 20%, and 5.6 mL·min^−1^), and 6 (60 g, 25%, and 5.6 mL·min^−1^), the MC of those samples were 4.15, 4.25, and 5.37%, respectively, which is slightly higher than those that had the same pre-wet mass (60 g) and low flow rate (4.0 mL·min^−1^) in combination. However, in both cases, the MC was still in the normal range (<6%). Statistically, the main effects (pre-wet mass, flow rate, and WPH concentration) and their interactions had significant (*p* < 0.05) effects on the MC. The MC was also significantly different (*p* < 0.05) among the replications.

The regression coefficients for the statistically significant models are given in Table 5. The MC model equation is Ŷ = 9.0147 − 0.2080x_1_ − 0.1828x_2_ + 0.5160x_3_ + 0.0007(x_1_)^2^ 0.0038x_1_x_2_ + 0.0319(x_2_)^2^ + 0.0131x_1_x_3_ − 0.0657x_2_x_3_.

The influence of the pre-wet mass, WPH concentration and flow rate on the MC is shown in Figure 4. The increase in pre-wet mass had highly influenced the MC Figure 4a,b. Flow rate had also increased the MC in agglomerated WPI, whereas WPH concentration slightly affected the MC in agglomerated WPI. A high pre-wet mass and high flow rate in these treatments caused a collapse in the fluid bed. This can be explained by the plasticization of the entire particle surface due to the increasing humidity of the air inside the fluid bed. Large clumps were formed and settled in the bottom of the fluid bed chamber (Figure 3) because the shear forces acting on the particles in the fluid bed are no longer sufficient to destroy the numerous sinter bridges generated, which leads to a rapidly progressing cake formation [34]. High MC in the powder may influence the shelf-life due to the Maillard reaction, lumps formation and microbial growth [35].

It was reported by Gaiani et al. [36] that agglomerated powder contains a slightly lower MC than non-agglomerated WPI. In this study, the non-agglomerated WPI sample had about a 3.61% MC; some of the agglomerated WPI samples contained a slightly lower MC than 3.61%. In contrast, the others did not, and the highest MC detected among the agglomerated samples was 14.79% (treatment 14) Table 4. This might be due to the differences in processing temperatures or the higher flow rate which followed in this study.

#### 3.3.2. Water Activity

Water activity was positively correlated with the MC of the powders. In agglomerated powders, the higher MC resulted in higher water activity (Table 4). Water activity was significantly different (*p* < 0.05) among the treatments. It was in the range of 0.10–0.67. As observed in the MC, water activity had the highest values in treatments 14, 16, and 18. These treatments had a combination of a high per-wet mass (140 g) and high flow rate (5.6 mL·min^−1^), and their water activity values were 0.67, 0.33, and 0.34, respectively. In these treatments, large and hard clumps were formed, which settled in the bottom of the fluid bed chamber (Figure 3). Clump formation made it difficult to monitor the temperature of the end product. This resulted in a large variation in water activity in these treatments. The rest of the agglomerated samples had a water activity of 0.20. All main effects and their interactions significantly (*p* < 0.05) influenced the water activity.

#### 3.3.3. Relative Dissolution Index (RDI)

The number of fine particles with a chord length of <10 µm was plotted vs. rehydration time. Figure 5 shows a typical FBRM plot obtained during the rehydration of agglomerated WPI with WPH as a binder. As shown in Figure 5, the initial increase in the fine particle count during the first 142–171 s of rehydration can be attributed to the breakage/dissociation of the agglomerated WPI particles. Subsequently, a gradual decrease in the fine particle count was observed at 319–348 s into the dissolution of the agglomerated WPI. As the dissolution continued further, the fine particle count decreased below 0.5 µm, which is below the FBRM detection limit specified by the manufacturer. It was interesting to observe a slight decrease between those two stages, followed by a slight increase in the fine particle count during dissolution in all agglomerated WPI. More studies are needed to understand the phenomena.

It was interesting to observe a striking contrast between the dissolution behavior of agglomerated WPI and milk protein concentrates (MPC), as observed from the fine particle count obtained from the FBRM. Hauser and Amamcharla [21] observed a continuous increase in the fine particle count during the dissolution of MPC 90. The fine particle count continued to increase until the first 900 s of MPC 90 dissolution and no further increase in the fine particle count was observed. On the other hand, the dissolution of agglomerated WPI followed a different trend, as shown in Figure 5, which may be due to the compositional differences between WPI (mostly whey proteins) and MPC (caseins and whey proteins) powder.

The mean RDI of the agglomerated WPI samples manufactured as per the experimental design was between 63.20 and 95.57% (Table 4). Treatment 4 (60 g, 20 %, 5.6 mL·min^−1^) had the highest RDI of 95.57%, followed by treatment 6 (60 g, 25%, 5.6 mL.min^−1^) and treatment 5 (60 g, 25%, 4.0 mL·min^−1^), that had RDIs of 86.56 and 84.12%, respectively. There were no significant differences (*p* > 0.05) among these treatments. Treatment 3 (60 g, 20%, 4 mL·min^−1^) had the lowest RDI of 63.20 among the resultant agglomerated samples, followed by treatment 15 (140 g, 20%, 4 mL·min^−1^), and treatment 1 (60 g, 15%, 4 mL·min^−1^) that had RDIs of 67.98 and 69.32%, respectively. There was also no significant difference (*p* > 0.05) among those treatments.

Pre-wet mass and flow rate had a significant effect (*p* < 0.05) on the RDI. WPH concentration did not have a significant (*p* > 0.05) difference as the main factor. However, its interactions with other factors were significantly different (*p* < 0.05). The interactions, pre-wet × WPH concentration, pre-wet × flow rate, WPH concentration × flow rate, and pre-wet × WPH concentration × flow rate, were all significantly (*p* < 0.05) different. In the case of 60 g and 140 g pre-wet mass, it was observed that a higher flow rate (5.6 mL·min^−1^) resulted in a higher RDI in agglomerated powders. On the other hand, a high flow rate resulted in a decrease in the RDI in treatments with 100 g pre-wet mass (Table 4). Vengateson and Mohan [37] reported that increasing the binder flow rate caused the formation of bigger granules with more flowability and decreased bulk density and friability, which consequently improved the rehydration.

The influence of pre-wet mass, WPH concentration, and flow rate on the RDI is shown in Figure 6. Figure 6a,b show that the increase in pre-wet mass had drastically decreased the RDI. Whereas the flow rate and WPH concentration positively influenced the RDI. As they increased, the RDI had increased, Figure 6a–c. The pre-wet mass and flow rate had significantly affected the RDI due to their influence on the structure and shape of the particles. The regression coefficients for the statistically significant models are given in Table 4. The RDI model equation is Ŷ= −10.3740 + 0.8782x_1_ − 2.5652x_2_ − 0.0201x_1_x_2_ (Table 4). Gaiani et al. [38] noted that the size associated with the shape descriptors could be an important factor influencing milk powders’ rehydration properties.

#### 3.3.4. Solubility Index (SI) 

SI measures the status of the powder after 30 min of rehydration, followed by centrifugation to remove the undissolved particles. In all agglomerated samples, SI was 95%. As expected, agglomeration processing conditions did not have any significant effect on the solubility index of the powders (*p* > 0.05). None of the processing main effects nor their interactions had significantly affected (*p* > 0.05). Chever et al. [39] reported also that agglomeration did not affect the solubility index of milk powders. 

#### 3.3.5. Emulsifying Capacity (EC)

The EC of agglomerated samples is shown in Table 4. It was within the range of 4.33–4.93 g of oil/mg of protein and there was no significant difference (*p* > 0.05) among the treatments. The highest EC was observed in treatment 7 (100 g, 15%, 4.0 mL·min^−1^). In contrast, the lowest EC was observed in treatment 6 (60 g, 25%, 5.6 mL·min^−1^). Pre-wet mass is the only main factor that had a significant effect (*p* < 0.05) on the EC. The EC was decreased with the increase in flow rate and WPH concentration. However, it was increased with the increase in the pre-wet mass. In the comparison of pre-wet levels, 60 and 100 g pre-wet mass were those that were significantly different (*p* < 0.05) from each other. Hence, there was no significant difference (*p* > 0.05) among the main effect’s interactions. Turgeon et al. [40] had used the same method in determining the EC of WPC. The EC was approximately 4.6 g of oil per mg of protein, which matched our results of WPI agglomerated samples.

## 4. Conclusions

WPH samples had similar chemical and physical properties, indicating consistent manufacturing conditions and processes. Agglomeration conditions, especially the pre-wet mass and the flow rate, have affected the moisture content, water activity, and the RDI of agglomerated samples, primarily. Some of the agglomerated samples have failed to meet the industrial and ADPI standards for dry whey products. The data suggests that the WPH concentration has more impact on the physical properties of the agglomerates. Finally, the hydrolysate can be used as a binder and an alternative to soy lecithin.

## Figures and Tables

**Figure 1 foods-11-01797-f001:**
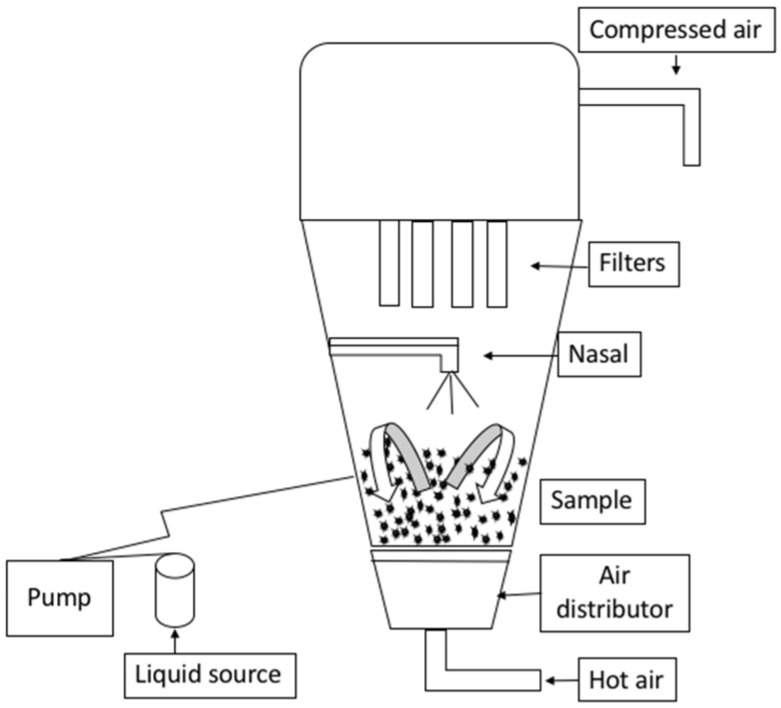
The top-spray fluid bed granulator (Midi-Glatt) that was used in agglomerating WPI powder.

**Figure 2 foods-11-01797-f002:**
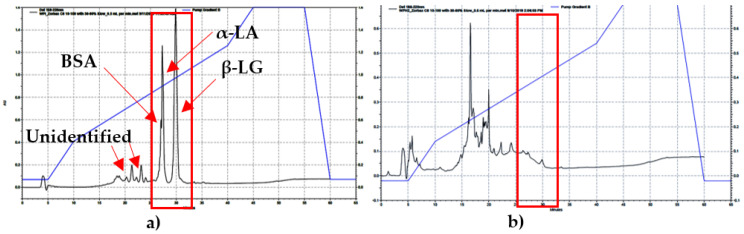
(**a**) The RP-HPLC chromatogram of WPI shows the major WPs in the unhydrolyzed sample; (**b**) The RP-HPLC chromatogram of WPH. No peaks were observed at the times of 26.3, 27.2, 29.9 min, indicating a complete hydrolysis of the major WPs in WPH samples.

**Figure 3 foods-11-01797-f003:**
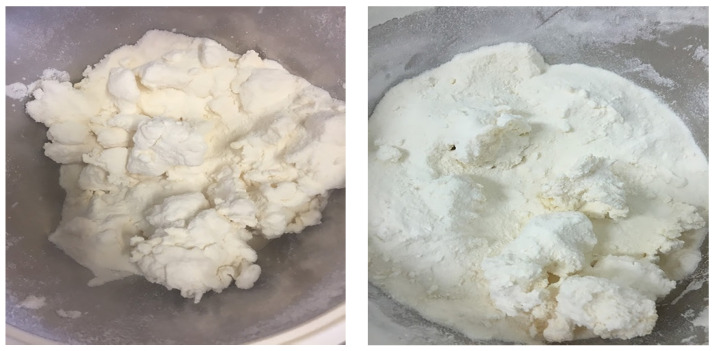
The resultant agglomerated powder in treatments that had the combination of 140 g pre-wet mass and 5.6 mL·min^−1^ flow rate. The resultant powder did not meet the industrial specification of WPI.

**Figure 4 foods-11-01797-f004:**
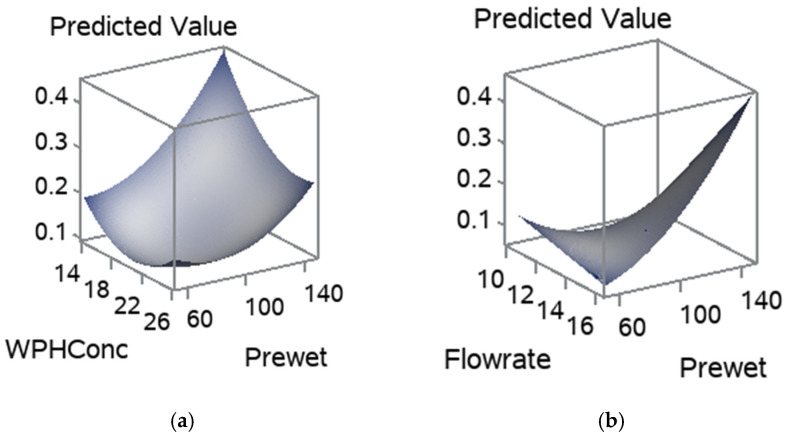
Response surface for MC, (**a**) pre-wet mass vs. WPH concentration; (**b**) flow rate and pre-wet mass; and (**c**) WPH concentration and flow rate.

**Figure 5 foods-11-01797-f005:**
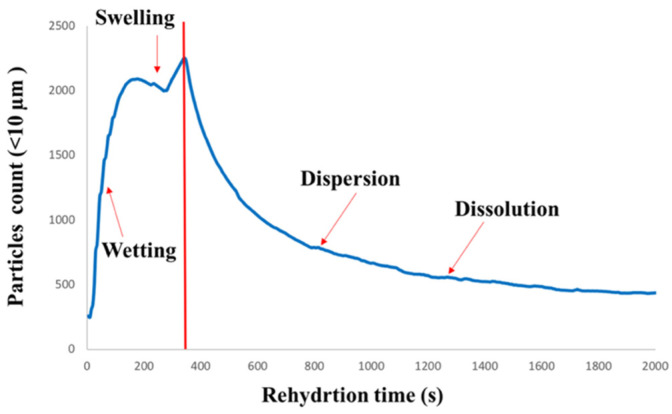
A typical fine particle count of chord length <10 µm plot vs. rehydration time. All four rehydration phases (wetting, swelling, dispersion, and dissolution) are identified on the figure as well.

**Figure 6 foods-11-01797-f006:**
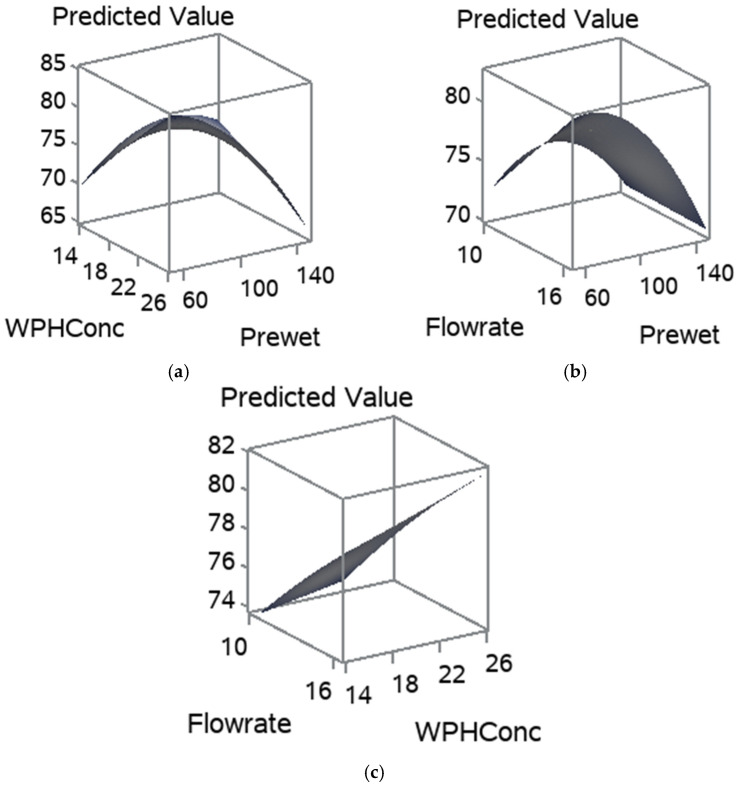
Response surfaces for RDI, for (**a**) pre-wet mass vs. WPH concentration; (**b**) flow rate and pre-wet mass; and (**c**) WPH concentration and flow rate.

**Table 1 foods-11-01797-t001:** Moisture content (%), protein content (%), and degree of hydrolysis of WPH and WPI samples.

Samples	Moisture Content (%)	Protein Content (%)	Degree of Hydrolysis (%)
WPH lot 1	2.99	91.96	14.80 ± 0.35
WPH lot 2	2.67	91.57	15.35 ± 0.64
WPH lot 3	2.88	92.28	13.82 ± 0.39
WPI	3.61	94.02	-

All values are expressed as mean ± SD, N = 3.

**Table 2 foods-11-01797-t002:** Water activity, mean particle size, and zeta potential of WPH and WPI sample.

Samples	Water Activity	Mean Particle Size (nm)	Zeta Potential (mV)
WPH lot 1	0.20 ± 0.02 ^ab^	181.23 ± 0.60 ^b^	−23.86 ± 2.03
WPH lot 2	0.21 ± 0.01 ^a^	150.67 ± 3.65 ^c^	−24.04 ± 1.40
WPH lot 3	0.16 ± 0.00 ^b^	198.93 ± 9.77 ^a^	−22.88 ± 1.16
WPI	0.12 ± 0.01	112.17 ± 14.40	−19.59 ± 1.58

^a–c^ Means with different superscripts are significantly different (*p* < 0.05). All values are expressed as mean ± SD, n = 3.

**Table 3 foods-11-01797-t003:** Bulk and tapped density, color of WP1 and WPH samples.

Samples	Bulk Density (g/cm^3^)	Tapped Density (g/cm^3^)	Color		
			L*	a*	b*
WPH lot 1	0.33 ± 0.00 ^a^	0.45 ± 0.00 ^a^	91.95 ± 0.83	0.15 ± 0.04 ^b^	7.49 ± 0.15 ^b^
WPH lot 2	0.31 ± 0.00 ^b^	0.42 ± 0.00 ^c^	92.39 ± 0.21	0.16 ± 0.01 ^b^	7.29 ± 0.15 ^b^
WPH lot 3	0.32 ± 0.01 ^ab^	0.44 ± 0.00 ^b^	92.11 ± 0.25	1.54 ± 0.03 ^a^	10.52 ± 0.11 ^a^
WPI	0.30 ± 0.00	0.42 ± 0.00	92.73 ± 0.49	1.22 ± 0.06	10.39 ± 0.04

^a–c^ Means with different superscripts are significantly different (*p* < 0.05). All values are expressed as mean ± SD, n = 3.

**Table 4 foods-11-01797-t004:** Moisture content (%), water activity, solubility index (%), emulsifying capacity (g of oil/mg of protein), and relative dissolution index of all agglomerated WPI treatments as per experimental design.

Treatment	Pre-Wet Mass (g)	WPH Concentration (%)	Flow Rate (mL·min^−1^)	Moisture Content (%)	Water Activity	Relative Dissolution Index (%)	Solubility Index (%)	Emulsifying Capacity (g of oil/mg of Protein)
1	60	15	4.0	3.36 ± 0.15 ^c^	0.13 ± 0.01 ^d^	69.324 ± 3.79 ^def^	95.36 ± 0.02	4.66 ± 0.29
2	60	15	5.6	5.37 ± 1.60 ^bc^	0.21 ± 0.08 ^cd^	73.37 ± 6.48 ^bcdef^	95.43 ± 0.08	4.58 ± 0.34
3	60	20	4.0	3.35 ± 0.52 ^c^	0.12 ± 0.01 ^d^	63.20 ± 0.94 ^f^	95.34 ± 0.03	4.59 ± 0.29
4	60	20	5.6	4.15 ± 0.16 ^c^	0.11 ± 0.01 ^d^	95.57 ± 4.32 ^a^	95.48 ± 0.19	4.40 ± 0.42
5	60	25	4.0	3.48 ± 0.52 ^c^	0.10 ± 0.02 ^d^	84.12 ± 5.02 ^abc^	95.33 ± 0.02	4.40 ± 0.52
6	60	25	5.6	4.25 ± 0.07 ^c^	0.13 ± 0.01 ^d^	86.56 ± 3.03 ^ab^	95.38 ± 0.01	4.33 ± 0.42
7	100	15	4.0	4.48 ± 0.73 ^c^	0.18 ± 0.00 ^d^	81.83 ± 3.24 ^abcd^	95.41 ± 0.05	4.93 ± 0.20
8	100	15	5.6	5.39 ± 0.37 ^bc^	0.21 ± 0.00 ^cd^	78.05 ± 0.67 ^bcdef^	95.45 ± 0.03	4.75 ± 0.11
9	100	20	4.0	4.26 ± 0.41 ^c^	0.16 ± 0.00 ^d^	82.83 ± 5.19 ^abcd^	95.41 ± 0.03	4.80 ± 0.13
10	100	20	5.6	3.77 ± 0.25^c^	0.14 ± 0.00 ^d^	75.25 ± 5.55 ^bcdef^	95.39 ± 0.00	4.82 ± 0.20
11	100	25	4.0	3.73 ± 0.45 ^c^	0.14 ± 0.02 ^d^	79.13 ± 1.56 ^bcde^	95.39 ± 0.02	4.73 ± 0.27
12	100	25	5.6	3.66 ± 0.16 ^c^	0.14 ± 0.03 ^d^	76.50 ± 6.62 ^bcdef^	95.39 ± 0.05	4.47 ± 0.55
13	140	15	4.0	3.30 ± 0.13 ^c^	0.12 ± 0.01 ^d^	74.19 ± 4.91 ^bcdef^	95.36 ± 0.01	4.63 ± 0.11
14	140	15	5.6	14.79 ± 4.50 ^a^	0.67 ± 0.10 ^a^	76.64 ± 6.17 ^bcdef^	95.94 ± 0.31	4.52 ± 0.30
15	140	20	4.0	3.51 ± 0.06 ^c^	0.12 ± 0.01 ^d^	67.98 ± 4.58 ^ef^	95.36 ± 0.05	4.75 ± 0.29
16	140	20	5.6	7.56 ± 1.32 ^b^	0.33 ± 0.07 ^bc^	75.30 ± 4.48 ^bcdef^	95.51 ± 0.07	4.82 ± 0.20
17	140	25	4.0	3.56 ± 0.10 ^c^	0.12 ± 0.01 ^d^	70.64 ± 4.18 ^cdef^	95.34 ± 0.04	4.76 ± 0.25
18	140	25	5.6	7.41 ± 3.46 ^b^	0.34 ± 0.10 ^b^	75.37 ± 10.42 ^bcdef^	95.51 ± 0.15	4.71 ± 0.39

^a–f^ Means with different superscripts are significantly different (*p* < 0.05). All values are expressed as mean ± SD. n = 3.

**Table 5 foods-11-01797-t005:** The coefficients of moisture content and relative dissolution rate models for the response variables.

Parameter	MC (%)	RDI (%)
Intercept (β_0_)	9.0147	−10.3740
Linear	*p* (<0.0001)	*p* (0.0011)
Pre-wet mass (β_1_)	−0.2080	0.8782
WPH Concentration (β_2_)	−0.1828	2.5652
Flow rate (β_3_)	0.5160	NS
Quadric	*p* (0.0028)	*p* (0.3129)
Pre-wet*Pre-wet (β_11_)	0.0007	NS
WPH Concentration*WPH Concentration (β_22_)	0.0319	NS
Cross product	*p* (<0.0001)	*p* (0.0039)
Pre-wet mass*WPH Concentration (β_12_)	−0.0038	−0.0202
Pre-wet mass*Flowrate (β_13_)	0.0131	NS
WPH Concentration*Flowrate (β_23_)	−0.0658	NS
*p* (Model)	<0.0001	0.0002

NS = not significant (*p* > 0.05).

## Data Availability

The datasets generated for this study are available on request to the corresponding author.

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
