# Peer review of "Characterization of a Commercial Whey Protein Hydrolysate and Its Use as a Binding Agent in the Whey Protein Isolate Agglomeration Process"

_foods, 2022, doi:10.3390/foods11121797_

Round 1
Reviewer 1 Report
This study was to characterize the chemical properties of three lots of whey protein hydrolysate and it was to evaluate the effectiveness as a binder in whey protein isolate wet agglomeration, it obtained from a commercial manufacturer.
The manuscript need corrections.
In table I the authors must put standard deviation.
In table 2 the authors must include WPI in the statistical analysis.
It Is very important put a table with the polynomial equation and regression coefficients of response variables
The figure 4 and 6 must be more big.
The conclusions
Delete We believe… (Line 534)…,functional properties (line 536).
Delete Therefor…. (Line 537), Line 538 and 539
Author Response
- In table I the authors must put standard deviation.
AU: The protein content and moisture content data was provided by the manufacturer and not available. Degree of hydrolysis was carried out by the authors in triplicate.
- Comment: In table 2 the authors must include WPI in the statistical analysis.
AU: The comparison is between the different lots of WPH and hence the comparison is only made between the WPH.
- Comment: It Is very important put a table with the polynomial equation and regression coefficients of response variables.
AU: The regression coefficients are provided both in Table 4 and also in text. Highlighted for your reference.
- The figure 4 and 6 must be more big.
AU: Modified as suggested.
- The conclusions
Delete We believe… (Line 534)…,functional properties (line 536).
Delete Therefor…. (Line 537), Line 538 and 539
AU: Modified as suggested.
Reviewer 2 Report
no
Author Response
None to modify
Reviewer 3 Report
The manuscript ‘Characterization of a commercial whey protein hydrolysate and its use as a binding agent in whey protein isolate agglomeration process’ has been reviewed and it’s a well-written manuscript. The authors are requested to consider the following suggestion for a better understanding of the research prospectus.
Suggestions:
- Authors may illustrate the manuscript concept in a flowchart format which may improve the presentation of the manuscript.
- Authors may include other important functional properties like non-protein-nitrogen, 5-hydroxymethyl-2- furfural, free-sulfhydryl levels if possible.
- Authors may discuss briefly about hydrolysis time, protein composition and enzymatic specificity in the manuscript as these functional efficiencies of WPHs is important in food applications.
- Authors are requested to mention Solubility index of WPHs at different pH levels for a better understanding.
- Authors are requested to mention the enzymes used for the whey protein hydrolysis for a clear understanding about the chemical behaviour of WPHs as binding agent.
Author Response
Authors may illustrate the manuscript concept in a flowchart format which may improve the presentation of the manuscript.
AU: Thank you for the suggestion. We think the section 2.1 along with the figure 1 should sufficiently describe the experimental design and the agglomeration process. It is excluded to avoid duplication of the material.
Authors may include other important functional properties like non-protein-nitrogen, 5-hydroxymethyl-2- furfural, free-sulfhydryl levels if possible.
AU: This is a very good suggestion. However, NPN and HMF are very helpful if the agglomeration causes denaturation of the proteins in WPI and heating above the denaturation temperatures are involved. Since the agglomeration was carried out below the denaturation temp, we did not include the analysis in the work.
*In our preliminary testing, we had performed NPN analysis and did not find any significant changes.
Authors may discuss briefly about hydrolysis time, protein composition and enzymatic specificity in the manuscript as these functional efficiencies of WPHs is important in food applications.
AU: Since the WPH samples were procured from a commercial source, it is proprietary.
Authors are requested to mention Solubility index of WPHs at different pH levels for a better understanding.
AU: The solubility at different pH levels is out of scope of this publication.
Authors are requested to mention the enzymes used for the whey protein hydrolysis for a clear understanding about the chemical behavior of WPHs as binding agent.
AU: Since the WPH samples were procured from a commercial source, it is proprietary information.
Round 2
Reviewer 1 Report
The authors did the corrections